# Targeting Peptide, Fluorescent Reagent Modified Magnetic Liposomes Coated with Rapamycin Target Early Atherosclerotic Plaque and Therapy

**DOI:** 10.3390/pharmaceutics14051083

**Published:** 2022-05-18

**Authors:** Chen Huang, Wentao Huang, Lifen Zhang, Chunyu Zhang, Chengqian Zhou, Wei Wei, Yongsheng Li, Quan Zhou, Wenli Chen, Yukuan Tang

**Affiliations:** 1Department of Minimally Invasive Interventional Radiology, Guangzhou Panyu Central Hospital, Guangzhou 511400, China; 05yxhc@163.com; 2MOE Key Laboratory of Laser Life Science & Institute of Laser Life Science, College of Biophotonics, South China Normal University, Guangzhou 510631, China; 2018022750@m.scnu.edu.cn (W.H.); 2019022756@m.scnu.edu.cn (L.Z.); 2019022755@m.scnu.edu.cn (C.Z.); chenwl@scnu.edu.cn (W.C.); 3Guangdong Provincial Key Laboratory of Laser Life Science, College of Biophotonics, South China Normal University, Guangzhou 510631, China; 4Institute for Brain Research and Rehabilitation, South China Normal University, Guangzhou 510631, China; 5Neuroscience Laboratory, Hugo Moser Research Institute at Kennedy Krieger, Baltimore, MD 21205, USA; czhou65@bu.edu; 6Institution of Guang Dong Cord Blood Bank, Guangzhou 510700, China; wwei@chinacord.org (W.W.); ysli@chinacord.org (Y.L.); 7Department of Radiology, The Third Affiliated Hospital of Southern Medical University, Guangzhou 510630, China

**Keywords:** early atherosclerosis, VCAM-1, MRI, rapamycin, bimodal imaging, Rap/Fe_3_O_4_@VHP-Lipo

## Abstract

Atherosclerosis is the leading cause of global morbidity and mortality. Its therapy requires research in several areas, such as diagnosis of early arteriosclerosis, improvement of the pharmacokinetics and bioavailability of rapamycin as its therapeutic agents. Here, we used the targeting peptide VHPKQHR (VHP) (or fluorescent reagent) to modify the phospholipid molecules to target vascular cell adhesion molecule-1 (VCAM-1) and loaded ultrasmall paramagnetic iron oxide (USPIO/Fe_3_O_4_) plus rapamycin (Rap) to Rap/Fe_3_O_4_@VHP-Lipo (VHPKQHR-modified magnetic liposomes coated with Rap). This nanoparticle can be used for both the diagnosis and therapy of early atherosclerosis. We designed both an ex vivo system with mouse aortic endothelial cells (MAECs) and an in vivo system with ApoE knockout mice to test the labeling and delivering potential of Rap/Fe_3_O_4_@VHP-Lipo with fluorescent microscopy, flow cytometry and MRI. Our results of MRI imaging and fluorescence imaging showed that the T2 relaxation time of the Rap/Fe_3_O_4_@VHP-Lipo group was reduced by 2.7 times and 1.5 times, and the fluorescence intensity increased by 3.4 times and 2.5 times, respectively, compared with the normal saline group and the control liposome treatment group. It showed that Rap/Fe_3_O_4_@VHP-Lipo realized the diagnosis of early AS. Additionally, our results showed that, compared with the normal saline and control liposomes treatment group, the aortic fluorescence intensity of the Rap/Fe_3_O_4_@VHP-Lipo treatment group was significantly weaker, and the T2 relaxation time was prolonged by 8.9 times and 2.0 times, indicating that the targeted diagnostic agent detected the least plaques in the Rap/Fe_3_O_4_@VHP-Lipo treatment group. Based on our results, the synthesized theragnostic Rap/Fe_3_O_4_@VHP-Lipo serves as a great label for both MRI and fluorescence bimodal imaging of atherosclerosis. It also has therapeutic effects for the early treatment of atherosclerosis, and it has great potential for early diagnosis and can achieve the same level of therapy with a lower dose of Rap.

## 1. Introduction

Atherosclerosis is a chronic cardiovascular disease caused by excessive accumulation of lipids, especially cholesterol, under the endothelium that arises in the local artery wall [1,2]. The early diagnosis and treatment of atherosclerosis are research hotspots for reducing the harm caused by atherosclerosis effectively. The formation and development of atherosclerotic plaques is quite complex and involves dyslipidemia, endothelial cells and immune dysfunction [3,4,5]. Among them, the early stages of the disease are characterized by vascular endothelial dysfunction. Leukocytes are recruited to the endothelium and express pro-inflammatory cytokines. Pro-inflammatory cytokines lead to up-regulated expression of vascular cell adhesion molecule-1 (VCAM-1) on the surface of endothelial cells [5,6,7]. Expression of the adhesion molecule VCAM-1 occurs early during atherogenesis and may, therefore, serve as a useful imaging biomarker of inflammation in atherosclerosis [8].

The active targeting agents commonly used are mainly peptides, anti-bodies, small molecules and proteins [9]. Medicines using polypeptides as targeted agents have the advantages of high stability, strong safety and low price. They have been extensively studied as an effective therapy for various diseases, including cancer, metabolic and autoimmune diseases [10]. The peptide Val-His-Pro-Lys-Gln-His-Arg (VHPKQHR, VHP) contains homology to very late antigen-4 (VLA-4), a known ligand of VCAM-1 [8,11] VCAM-1 overexpression that plays a vital role in the development of atherosclerosis and can be used as a key target for early atherosclerosis [8], but few studies have used it as a target receptor for early plaque diagnostic combination therapy.

Rapamycin (Rap) with 5 μM (4.57 μg/mL) inhibits the formation of foam cells (which is an early sign of atherosclerosis) by blocking activation of the mammalian target of rapamycin (mTOR). Although increasing evidence suggests that rapamycin therapy leads to side effects, such as dyslipidemia and insulin resistance, the net positive effect still outweighs its disadvantages. However, rapamycin’s immunosuppressive activity and strong hydrophobicity limit its clinical application [12].

Liposomes are self-assembled structures made up of lipid bilayers. The lipid bilayers completely surround the water core and can transport various types of hydrophobic active substances and biomolecules [13]. Liposome drug delivery systems have been made into anti-tumor drugs [14,15], anti-cardio-cerebrovascular disease drugs, [16] antipsychotic drugs [17], antibacterial drugs [18] and vaccines [19] with significant progress, showing slow and controllable release, good biocompatibility and low toxicity and side effects. The targeting peptide VHPKQHR (VHP) (or fluorescent reagent) to modify the phospholipid molecules has not been reported for the synthesis of liposome nanoparticles.

The main methods used in the diagnosis of atherosclerosis are computed tomography (CT) [20], positron emission tomography (PET) [21], ultrasound [22] and magnetic resonance imaging (MRI) [23]. Among these, MRI is a powerful technique for imaging atherosclerotic plaque morphology and is becoming more widely used due to its non-injury and high soft tissue resolution [24]. However, since the morphology of the early atherosclerotic lesion area is not obviously stenotic and, therefore, not easily detectable, improving the contrast ratio of specific tissues to surrounding tissues would be beneficial to solve this problem. Therefore, it would be extremely useful to design MRI contrast agents that selectively bind to plaque receptors for accurate plaque diagnosis. The addition of a large amount of aggregated contrast agent reduces the relaxation time of the lesion part, thereby achieving significant contrast enhancement. The ultra-high field of view T2 MRI has gradually aroused the demand for biomaterial research and clinical diagnosis due to its high resolution and high detection accuracy [25]. Ultra-small superparamagnetic iron oxide (USPIO) is a new MRI contrast medium, mainly producing a T2 negative contrast effect [26]. It can be designed in different sizes according to different needs and has been clinically approved in the United States and Europe, making it very user-friendly and safe. In addition, compared to a single non-invasive imaging technology, multimodal imaging is a more accurate method for plaque diagnosis and has become a hot topic in cardiovascular disease diagnosis in recent years [24]. According to recent studies, fluorescence imaging is rarely used to study the status of atherosclerotic lesions, so combining MR and fluorescence imaging to monitor the atherosclerotic lesions would be a new and attractive strategy. 

In this study, we synthesized a theragnostic liposome with great potential for clinical application. It not only wraps contrast agent USPIO (Fe_3_O_4_) and low-dose therapeutic drugs Rap but also connects to target the VHPKQHR (VHP) peptide at the plaque and fluorescent reagent for the diagnosis and therapy of early atherosclerotic plaque in mice. The principles of magnetic liposomes synthesis and targeted therapy are shown in Figure 1. Our hypothesis: the targeted Rap-loaded liposome is first identified by VCAM-1 at the plaque and then enters endothelial cells, macrophages and vascular smooth muscle cells in the plaque through endocytosis. Due to damage to the endothelial layer, the permeability of the vascular wall is increased and liposomes can easily penetrate the plaque through the gap in the destroyed endothelial layer [27]. The Rap in the plaque is released slowly from liposomes in the slightly acidic environment. There are many foam cells evolved from macrophages inside the plaque [28], and the foam cells engulf the liposomes; then, the Rap encapsulated in the liposomes is released and degrades the lipids in the foam cells, thus reducing the oxidized LDL. 

## 2. Materials and Methods

All animal procedures were performed in accordance with National Institutes of Health (NIH) guidelines and laboratory animal guidelines of South China Normal University, and the experiments were approved by Animal Ethics Committee of South China Normal University.

### 2.1. Synthesis of Liposomes

#### 2.1.1. Drug information

DSPE-MPEG_2000_ (N- (Carbonyl-methoxypolyethylene glycol 2000)-1,2-distearoyl-sn-glycerol-3-phosphoethanolamine), DSPE-PEG_2000_(1,2-distearoyl-sn–glycero-3-phosphoethanolamine-poly (ethylene glycol))-NHS, DSPE-fluoresceine isothiocyanate (DSPE-FITC) were offered by Xi’an Ruixi Biological Technology Co., Ltd. (Xi’an, China) Dimethyl Formamide (DMF) and N-methyl Morpholine were purchased from Sigma-Aldrich, St. Louis, MO, USA. Peptide VHPKQHR (VHP) was obtained from Shanghai Taopu Biological Technology Co., Ltd. (Shanghai, China).

#### 2.1.2. Synthesis of Rap/Fe_3_O_4_@Lipo

90 mg Bovine serum albumin (BSA) was added to the USPIO solution [29] (10 mg USPIO, 10 mL ultrapure water) and activated by vigorous stirring under ice bath for 6 h. The obtained USPIO–BSA solution (1 mg/mL) was stored at 4 °C for subsequent experiments. Liposomes used in this study were prepared via the thin-film hydration method. Weighed 2 mg rapamycin (Bide Pharmatech Ltd., Shanghai, China), 10 mg egg yolk lecithin, 5 mg DSPE-MPEG_2000_ and 1 mg DSPE-FITC and dissolved them in 5 mL methanol and chloroform (Sigma-Aldrich, St. Louis, MO, USA) mixed solvent (1:1, *v*/*v*). The solvent was removed by 37 °C rotary evaporation under reduced pressure, and 2 mL USPIO–BSA was added for hydration; then, the obtained aqueous solution was dispersed by ultrasonic to obtain liposomes solution. Finally, centrifuge for 15 min to remove the solution through ultrafiltration centrifuge tube (300 kDa) (Corning Incorporated, Corning, NY, USA) at 6000 rpm, and redisperse in 2 mL pure water to obtain 1 mg/mL Rapamycin/Fe_3_O_4_@Liposome (non-targeting liposome, Rap/Fe_3_O_4_@Lipo, R/F@L).

#### 2.1.3. Synthesis of DSPE-PEG_2000_-VHP

200 mg DSPE-PEG_2000_-NHS was dissolved in 5 mL DMF, and then VHP (220 mg) and N-methyl Morpholine (220 mg) were added and the reaction mixture was stirred for 12 h at room temperature until it became homogeneous. The reaction solution was placed in a dialysis bag (molecular weight cutoff (MWCO) = 3500) and dialyzed in pure water for 24 h. The dialysate was collected and freeze-dried to obtain DSPE-PEG_2000_-VHP.

#### 2.1.4. Synthesis of Rap/Fe_3_O_4_@VHP-Lipo

2 mg rapamycin, 10 mg egg yolk lecithin, 5 mg DSPE-MPEG_2000_, 1 mg DSPE-PEG_2000_-VHP and 1 mg DSPE-FITC were dissolved in methanol and chloroform mixed solvent (1:1, *v*/*v*). The solvent was removed by rotary evaporation under reduced pressure at 37 °C, and then 2 mL USPIO–BSA was added for hydration. Further, the obtained aqueous solution was dispersed by ultrasonic to obtain liposomes solution. Finally, centrifuge for 15 min to remove the solution through ultrafiltration centrifuge tube (300 kDa) at 6000 rpm and redisperse in 2 mL pure water to obtain 1 mg/mL rapamycin/Fe_3_O_4_@VHPKQHR-Liposome (targeting liposome, Rap/Fe_3_O_4_@VHP-Lipo, R/F@V-L).

### 2.2. Characterization of Nanoparticles

Diameter and zeta potential of liposomes before and after attachment of peptides were measured by dynamic light scattering (DLS) using Malvern Zetasizer Nano ZS90 (Malvern Instruments, Malvern, UK). Transmission electron microscopy (TEM) (JEM-1400 PLUS, Tokyo, Japan) was used to observe the shape and true size of Rap/Fe_3_O_4_@VHP-Lipo. The inclusion of VHPKQHR and rapamycin in liposomes was examined by UV-Vis (ultraviolet visible spectroscopy) (Shimadzu, Kyoto, Japan). To test whether the reaction of VHP with DSPE-PEG_2000_-NHS was successful, the two individual substances and the end products of their reaction were analyzed by hydrogen-1 nuclear magnetic resonance (^1^H NMR) spectroscopy to determine them separately.

### 2.3. Drug Loading and Release

To measure the Rap/Fe_3_O_4_@VHP-Lipo encapsulation efficiency (EE) and loading efficiency (LE), the samples were analyzed by high-performance liquid chromatography (HPLC) (Shimadzu, Kyoto, Japan). A C18 column (4.6 × 250 mm, 5 μm particle size, Diamonsil) was used for the chromatographic separation. Rap/Fe_3_O_4_@VHP-Lipo was eluted under isocratic conditions using a binary solvent system [H_2_O + Methanol, 15:85 *v*/*v*] pumped at a flow rate of 1.0 mL/min. The ultraviolet (UV) detection was set at 277 nm. 

200 μL Rap/Fe_3_O_4_@VHP-Lipo solution was dissolved in HPLC-grade methanol (10 mL) and analyzed by LCMS. EE and LE were determined using the following equations:$$\mathrm{EE}=\frac{\mathrm{Measured}\;\mathrm{concentration}}{\mathrm{Theoretical}\;\mathrm{concentration}}\times 100\%$$
$$\mathrm{LE}=\frac{\mathrm{The}\;\mathrm{amount}\;\mathrm{of}\;\mathrm{rapamycin}\;\mathrm{measured}}{\mathrm{Amount}\;\mathrm{of}\;\mathrm{all}\;\mathrm{drugs}}\times 100\%$$

In vitro release investigation: the prepared liposomes were diluted to 100 μg/mL (rapamycin concentration), 1 mL of liposomal solution was packed into dialysis bags (molecular weight 8000–14,000), placed in 10 mL of PBS release medium with pH = 5.6 and shaken at 37 °C, 100 r/min. Then, all the release medium was removed at 1, 3, 5, 7, 12 and 24 h for concentration measuring and replenished with fresh release medium. The results are shown in Appendix A and Figure 1f. 

The amount of rapamycin in PBS was determined by high-performance liquid chromatography (HPLC) at 1, 3, 5, 7, 12 and 24 h (calculated based on the measured rapamycin concentration peak area regression line Y = 72122X − 8522.8, R^2^ = 0.9997). The release rate of rapamycin in vitro was calculated according to formula:$$\mathrm{Release}\;\mathrm{rate}\;\left(\%\right)=\frac{\mathrm{amount}\;\mathrm{of}\;\mathrm{free}\;\mathrm{rapamycin}}{\mathrm{amount}\;\mathrm{of}\;\mathrm{total}\;\mathrm{rapamycin}}\times 100\%$$

### 2.4. Cell Culture and Viability Assay

Mouse aortic endothelial cells (MAECs) were purchased from CHI Scientific; the cells were cultured in DMEM (high glucose) medium (Jiangsu KeyGEN BioTECH Corp., Ltd., Nanjing, China) with 10% fetal bovine serum (FBS) (Gibco, South America) solution in 5% CO_2_ atmosphere at 37 ℃. Cell Counting Kit-8 (CCK-8, Dojindo Laboratories, Kumamoto, Japan) colorimetric assays were performed to detect cell viability of MAECs. MAECs were cultured in 96-well plates (Corning Incorporated, Corning, NY, USA) at 2 × 104 cells per well under various concentrations (0, 10, 20, 30, 40, 60 μg/mL) of Rap/Fe_3_O_4_@ VHP-Lipo for 24 h. After that, the viability of cells was estimated by measurement of absorbance at 450 nm (A450), which was read with a microplate reader (INFINITE M200, Tecan, Männedorf, Switzerland).

### 2.5. Western Blot

In order to simulate the lipid-rich and high-inflammatory environment of endothelial cells in the plaque in vitro, lipopolysaccharides (LPS) (Sigma, Darmstadt, Germany) or tumor necrosis factor-α (TNF-α) (Sigma, Darmstadt, Germany) were used to stimulate MAECs [5,30], in this study: TNF-α-treated MAECs at concentrations of 10 ng/mL and 100 ng/mL, and LPS at concentrations of 0.5 μg/mL and 1 μg/mL for 24 h, respectively. Then, the cells were separated by cells spatulas and lysed with cell lysate solution for 45 min. The supernatant was collected after centrifugation at 7000 g for 10 min. 10% sodium dodecyl sulfate polyacrylamide gel electrophoresis (SDS-PAGE) was used to separate protein and then transferred to polyvinylidene fluoride (PVDF) membranes (Millipore, Bedford, MA, USA), which were incubated with the Rabbit Anti-VCAM-1 Polyclonal Antibody (Absin Bioscience Inc., Shanghai, China) (all at 1:1500 in TBST solution (150 mM NaCl, 10 mM Tris-HCl, pH 7.4, 0.1% Tween 20) containing 5% skim milk) overnight at 4 °C before incubation with secondary antibody (Goat Anti-Rabbit IgG (corresponding to VCAM-1) (at 1:3000) for 1 h. GAPDH (glyceraldehyde-3-phosphate dehydrogenase) was used as loading control. Signals were detected using an image analyzer (Tanon-5200, Shanghai, China). Immunoreactivity was detected with enhanced chemiluminescence and quantified using Image J software.

### 2.6. Flow Cytometry Analysis

MAECs were placed in 6-well plates. After the cell density reaches 70–80%, the cells were incubated with 40 μg/mL Rap/Fe_3_O_4_@Lipo, Rap/Fe_3_O_4_@VHP-Lipo and PBS in each well for 3, 6 h, respectively. The cells were washed three times with PBS, digested with trypsin, and finally analyzed by flow cytometer (CytoFLEX, Beckman Coulter, Brea, CA, USA).

### 2.7. Confocal Microscopy Imaging

The cellular uptake of Rap/Fe_3_O_4_@VHP-Lipo was determined by laser-scanning confocal fluorescence microscopy. MAECs dispersed in confocal dishes at 1 × 106 cells/well density and incubated cells with 10 ng/mL TNF-α for 24 h. When the cells’ density reached 70–80%, 40 μg/mL of Rap/Fe_3_O_4_@Lipo or Rap/Fe_3_O_4_@VHP-Lipo was introduced into the confocal dishes. After an incubation time of 1 h and 3 h, cells were washed three times with PBS and fixed with 4% paraformaldehyde (Biosharp, Guangzhou, China) for 30 min and incubated with 4 ℃, 4,6-diamidino-2-phenylindole (DAPI) (Genview, Houston, TX, USA) for a nuclear stain for 15 min. Finally, the cells were washed three times after DAPI solution was removed. The cell samples were examined by confocal microscopy (ZEISS LSM 880, Zeiss, Jena, Germany) at a 10 objective using 488 nm diode laser as excitation source and 525nm as the emission wavelength of FITC.

### 2.8. Establishment and Therapy of Atherosclerosis Model Mice

ApoE^−/−^ mice (Nanjing Biomedical Research Institute, Nanjing, China), aged 8 weeks, were fed with high-fat diet (HFD) containing 1% cholesterol, 15% lard, 0.25% sodium cholate, 5% yolk powder, 2.5% sugar and basal diet for 4 months. Mice were dwelled at 22 ± 2 ℃, 55 ± 5% relative humidity. 

Thirty of these mice fed a HFD for 2 months were divided into 3 groups for tail vein injection. The mice in these three groups were injected with normal saline, Rap/Fe_3_O_4_@Lipo and Rap/Fe_3_O_4_@VHP-Lipo, respectively, every 3 days and were treated for a total of 2 months. During the therapy, they continued to be fed HFD.

### 2.9. Magnetic Resonance Imaging

The T2 relaxation time of plaques were tested by 7.0 T superconductor experimental system T2 mapping sequence imaging (Pharma Scan70/60 US 7.0 T, Bruker Biospin MRI GmbH, Ettlingen, Germany) after injecting 24 h with normal saline, Rap/Fe_3_O_4_@Lipo and Rap/Fe_3_O_4_@VHP-Lipo (2 mg Fe_3_O_4_/kg body weight) separately; the Fe_3_O_4_ (USPIO) concentration used for tail vein injection was 1 mg/mL. The following parameters were used for the T2 mapping imaging: repetition time [TR]/echo time [TE] 2200/9.5 ms, field of view 2.3 cm × 2.3 cm, matrix 256 × 256, slice thickness 0.8 mm.

The therapeutic effects of Rap/Fe_3_O_4_@VHP-Lipo on plaques were detected by 9.4 T superconducting magnet system T_2_ weighted sequence imaging (BioSpec 94/30 USR 9.4T, Bruker, Ettlingen, Germany) after 2 months (injected every 3 days) treating with normal saline, Rap/Fe_3_O_4_@Lipo and Rap/Fe_3_O_4_@VHP-Lipo (2 mg rapamycin/kg body weight) separately; the package capacity of rapamycin in Rap/Fe_3_O_4_@Lipo and Rap/Fe_3_O_4_@VHP-Lipo is 1 mg/mL. The following parameters were used for the T_2_ weighted imaging: repetition time [TR]/echo time [TE] 1032/23 ms, field of view 3 × 3 cm, matrix 196 × 196, layer thickness 0.7 mm, layer spacing 0.3 mm.

In order to determine the transverse relaxation rate (r_2_) of magnetic liposomes, Rap/Fe_3_O_4_@Lipo and Rap/Fe_3_O_4_@VHP-Lipo were placed in 1.5mL Eppendorf tube at different concentration levels (0, 10, 20, 30, 40, 60 μg/mL), respectively, and performed T_2_ mapping sequence scan on a Philips 3.0 T Achieva MRI system (Healthcare, Best, The Netherlands). The imaging parameters are as follows: field of view (FOV) 100 mm, matrix 168 × 168, layer thickness 2 mm, layer spacing 0.2 mm, echo time (TE) 60 ms, repetition time (TR) 2 s. The region of interest (ROI) is the average of the three middle layers, and the transverse relaxation rate is obtained by fitting the linear relationship between the 1/T2 value and the Fe_3_O_4_ concentration.

### 2.10. Fluorescence Imaging

The targeting capability of Rap/Fe_3_O_4_@VHP-Lipo was measured by ex vivo and in vivo fluorescence imaging. Six ApoE^−/−^ plaque mice fed two-month HFD were tail-vein-injected with normal saline, Rap/Fe_3_O_4_@Lipo and Rap/Fe_3_O_4_@VHP-Lipo after 0 h, 24 h and 48 h for multimodal fluorescence imaging in mice to test the distribution of fluorescence in mice. Then, we isolated the aortic vessels of the mice injected 48 h after fluorescence imaging in vivo and carefully dissected the fatty tissue around the blood vessels. The fluorescence intensity of blood vessels was measured using a multi-mode small animal live imaging system (FX Pro, Bruker, Billerica, MA, USA) with 470 nm excitation wavelength and 535 nm emission wavelength.

### 2.11. Histological Examination

Hematoxylin and eosin (H&E) staining was used to determine the presence of plaque and tissue toxicity of liposomes. After the blood vessels and other organs (heart, liver, spleen, lung, kidney) were carefully taken from the anesthetized ApoE^−/−^ mice, the tissues were soaked in formalin for 24 h. The samples were dried by gradient alcohol, embedded in the embedding machine and cut into sections (4 μm) with a cryostat. The sections were serially collected on gelatin-coated glass slides and stained with H&E staining. Then, the H&E-stained sections were observed under a microscope (Mshot, Guangzhou Ming-mei Technology Co., Ltd., Guangzhou, China).

### 2.12. Pharmacokinetic Test of Rapamycin in Mice

Triple four-stage liquid chromatography–mass spectrometry (LC–MS) instrument (Prelude SPLC+TSQ Quantitative LC–MS/MS, Thermo Fisher Scientific, Waltham, MA, USA) was used to determine the pharmacokinetic curve of rapamycin in mice. A Thermo Hypersil Gold chromatographic column (Shimadzu, Kyoto, Japan, 50 mm × 2.1 mm, 1.9 μm particles) was used for separation by using water phase (0.1% formic acid, solvent A) and methanol phase (solvent B) as the mobile phase under gradient elution as follows: initial, 70% A; 0.00–1.00 min, 70% A; 1.00–2.50 min, 15% A; 2.50–3.00 min, 10% A; 3.00–5.00 min, 70% A. The flow rate was 0.3 mol·min-1 with an injection volume of 5 μL. The column oven was set to 40 °C. 

The H-ESI (heated-electrospray ionization source) positive electrospray ionization mode was used for ionization. The temperature and ion spray voltage were 300 ℃ and 3000 V, respectively. The sheath gas (Arb), auxiliary gas (Arb), and sweep gas (Arb) were at 40 psi, 8 psi and 0 psi, respectively. For each analyte, selected-reaction monitoring (SRM) was selected. For each internal standard, only a quantifier SRM is needed. High purity argon was used as the sheath gas, auxiliary gas and collision gas. 

The peak areas of rapamycin at concentrations of 0.1, 1, 5, 10, 25, 50, 100 ng/mL were determined by LC–MS, and the regression line (Y = 655.4X − 225.18, R^2^ = 0.995) between the concentration and the peak area was drawn. 

Take 9 mice fed with HFD 2 months, divide them into 3 groups, 3 mice in each group, fast for 12 h before administration and drink freely. Mice in each group were injected with targeting liposomes, non-targeted liposomes and physiological saline, respectively, through the tail vein. At 1 h, 3 h, 6 h, 12 h, 24 h after administration, 20 μL of blood was collected from the tip of the tail, placed in a sodium heparin centrifuge tube, 80 μL of cell lysate was added and 400 μL of methanol was added to precipitate the protein and centrifuged at 4 °C 12,000 rpm (10,142× *g*) for 15 min, taking 500 μL of the supernatant. 100 μL methanol solution was reconstituted and centrifuged at 12,000 rpm (10,142× *g*) for 5 min; the supernatant was taken and filtered with a 0.22 μm filter membrane before testing. The result of sample loading is substituted into the regression line equation that has been measured to obtain the concentration of rapamycin.

### 2.13. Blood Biochemical Analysis

Fresh blood obtained from treated mice was dealt with static settlement at room temperature for 30 min and then centrifuged at 650 g for 10 min at 4 ℃. The supernatant was collected and sent to South China Normal University Hospital (Guangzhou, China) for examination of triglycerides (TG), total cholesterol (TC), low-density lipoprotein cholesterol (LDL-C), high-density lipoprotein cholesterol (HDL-C), blood urea nitrogen (UREA), creatinine (CREA), uric acid (UA) and aspartate transaminase (AST).

### 2.14. Data Analysis

GraphPad Prism 7.0 software (GraphPad Software, La Jolla, CA, USA) was used for statistical analysis. Experimental data were presented as mean ± standard deviation (SD). Two-tailed *t*-test was used between the two groups, multiple comparisons were used between multiple groups and differences between groups were analyzed using ANOVA. *p* < 0.05 was considered to be statistically significant.

## 3. Results

### 3.1. Synthesis and Characterization of Rap/Fe_3_O_4_@VHP-Lipo

First, DSPE-PEG2000(1,2-distearoyl-sn–glycero-3-phosphoethanolamine-poly (ethylene glycol))-NHS and VHP were dissolved in DMF (dimethyl formamide), and then they reacted under the catalysis of N-methyl morpholine to obtain the peptide-linked phospholipid molecule DSPE-PEG2000-VHP (Appendix A). Rap, USPIO–BSA (Fe_3_O_4_), egg yolk lecithin, DSPE-FITC (DSPE-fluoresceine isothiocyanate) (Appendix A) and DSPE-MPEG2000 (N- (Carbonyl-methoxypolyethylene glycol 2000)-1,2-distearoyl-sn-glycerol-3-phosphoethanolamine) (Appendix A) were dissolved and mixed with the DSPE-PEG2000-VHP to obtain liposomes (Lipo) Rap/Fe_3_O_4_@VHP-Lipo solution by the film dispersion method. After the reaction, DSPE-PEG_2000_-NHS, VHP and DSPE-PEG_2000_-VHP were analyzed by hydrogen-1 nuclear magnetic resonance (^1^H NMR) spectroscopy and it was found that the reaction product DSPE-PEG_2000_-VHP possessed the hydrogen spectral peaks of DSPE-PEG_2000_-NHS and VHP, and the peak at 7 to 9 ppm was the peak on the heterocyclic ring of the peptide molecule (Appendix A). As shown in the structure diagram of Rap/Fe_3_O_4_@VHP-Lipo (Figure 1a), by such synthesis, Rap/Fe_3_O_4_@VHP-Lipo encapsulated not only in their hydrophilic cavities with USPIO–BSA (Fe_3_O_4_ nanoparticles, Appendix A) but also Rap at the hydrophobic intercontact end of phospholipid molecules, and the surface of the liposome was also connected with the targeting peptide VHP and fluorescent group FITC. Observation of the synthesized Rap/Fe_3_O_4_@VHP-Lipo under TEM showed that they were basically round with regular shapes, and most of the particle sizes were distributed between 60 and 120 nm (Figure 1b). The hydrated particle size of Rap/Fe_3_O_4_@Lipo (refers to liposomes without targeting peptide DSPE-PEG_2000_-VHP) measured by DLS (dynamic light scattering) was 160 ± 3 nm and the zeta potential was −8.3 ± 0.3 mV; after modifying the VHP peptide, the size changed into 246 ± 14 nm (Figure 1c) and the zeta potential turned into +4.3 ± 0.2 mV (Figure 1d), indicating that the peptide (positive charge) was modified successfully. As shown by the UV-Vis (ultraviolet visible spectroscopy) result (Figure 1e), pure VHP peptide had a characteristic absorption peak at 218 nm, pure Rap had a characteristic absorption peak at 269 nm, Rap/Fe_3_O_4_@Lipo liposomes had only one peak at 274 nm belonging to rapamycin, while Rap/Fe_3_O_4_@VHP-Lipo had two peaks; one is the peak of VHP at 206nm and the other is the peak of Rap at 279 nm, which suggested that VHP and Rap were successfully incorporated with targeting liposomes Rap/Fe_3_O_4_@VHP-Lipo in the experiment of measuring the release efficiency of Rap in Rap/Fe_3_O_4_@VHP-Lipo by HPLC. The release amounts of Rap measured at each time point were obtained according to the linear relationship (Y = 72122X − 8522.8, R2 = 0.9997) between the concentration of Rap standard and the peak area (Appendix A). The results showed Rap in 20% ethanol was logarithmically rising with the lapse of time, and the release efficiency reached 70% at 24 h (Figure 1f).

### 3.2. Toxicity of Rap/Fe_3_O_4_@VHP-Lipo

We used MAECs (mouse aortic endothelial cells) to test the Rap/Fe_3_O_4_@VHP-Lipo toxicity in vitro. CCK-8 (Cell Counting Kit-8) was used to assess the cell viability after incubation with the Rap/Fe_3_O_4_@VHP-Lipo (1 mg/mL) at different concentrations (0, 10, 20, 30, 40, 60 μg/mL) for 12 h. The results demonstrated that the viability of MAECs was not affected until the concentration reached 60 μg/mL (Figure 2a). We chose the concentration at which Rap/Fe_3_O_4_@VHP-Lipo did not produce obvious cytotoxicity (the cell viability was greater than 80% at 12 h) as the working concentration of Rap/Fe_3_O_4_@VHP-Lipo. Therefore, we chose 30 μg/mL Rap/Fe_3_O_4_@VHP-Lipo for processing the MAECs in the following cell experiments. Then, the toxicity of the Rap/Fe_3_O_4_@VHP-Lipo in vivo was estimated by blood biochemical analysis and morphological analysis of major organs after tail vein injection with 30 μg/mL for 15 days. Blood biochemical indexes UA (uric acid) (Figure 2b), AST (aspartate transaminase, an indicator of liver function) (Figure 2c), CREA (creatinine) (Figure 2d) and UREA (blood urea nitrogen, indicators of kidney function) (Figure 2e) in the experimental group had no significant statistical difference compared with the normal-saline-treated group. Further, the H&E-stained section results also showed that the two groups had no significant morphological differences in the organs heart, liver, spleen, lungs and kidneys (Figure 2f). The above results demonstrated that Rap/Fe_3_O_4_@VHP-Lipo had no significant biological toxicity in vivo.

### 3.3. Toxicity of Rap/Fe_3_O_4_@VHP-Lipo

MAECs-expressed high levels of VCAM-1 are characteristic of cells at the site of atherosclerosis lesions [31]. MAECs were stimulated to express VCAM-1 using the inflammatory cytokines TNF-α (tumor necrosis factor-α) or LPS (lipopolysaccharides), which have been reported to play a role [32,33]. Our results showed that MAECs stimulated with TNF-α overexpressed protein VCAM-1, but there was no overexpression in the LPS-stimulated group (Figure 3a). The quantitative analysis showed that the VCAM-1 protein expression levels in TNF-α-stimulated groups were increased by 7 to 11 times compared with tha in control group (Figure 3b) because the expression of VCAM-1 protein with 10 ng/mL TNF-α treatment had been significantly up-regulated. We used the concentration of 10 ng/mL TNF-α treating MAECs in the following cell-targeting experiments. The FITC fluorescence intensities of the stimulated MAECs after incubating with two kinds of Rap/Fe_3_O_4_@Lipo and Rap/Fe_3_O_4_@VHP-Lipo liposomes were observed via a confocal microscope. The results showed that some of the Rap/Fe_3_O_4_@VHP-Lipo adhered to the MAECs or was uptaken by MAECs, while the Rap/Fe_3_O_4_@Lipo hardly showed binding to stimulated MAECs (Figure 3c). The quantitative fluorescence statistics showed that the Rap/Fe_3_O_4_@VHP-Lipo treatment group had 2.4 times and 3.5 times more fluorescence than the Rap/Fe_3_O_4_@Lipo treatment group at 1 h and 3 h, respectively (Figure 3d). 

In addition, we also used the flow cytometry to analyze the FITC fluorescence intensity of the stimulated MAECs after incubating with Rap/Fe_3_O_4_@VHP-Lipo and Rap/Fe_3_O_4_@Lipo for 0, 3, 6 or 12 h, respectively. The results showed that fluorescence intensity increased with processing time in two groups, but the percentage of cells that reached the set fluorescence threshold was 53.00%, 69.30% and 76.41% in the Rap/Fe_3_O_4_@VHP-Lipo group, respectively, which obviously has stronger adhesion to MAECs relative to the control, with 11.25%, 26.43% and 52.41% in the Rap/Fe_3_O_4_@Lipo group (Figure 3e). These results showed that Rap/Fe_3_O_4_@VHP-Lipo has a stronger targeting ability to MAECs of overexpression VCAM-1.

### 3.4. Targeting Effect and Prolonged Circulation Time of Rap/Fe_3_O_4_@VHP-Lipo In Vivo

After confirming that Rap/Fe_3_O_4_@VHP-Lipo was biologically safe and could target to the MAECs-overexpressed VCAM-1, we carried out the next in vivo targeting experiment. ApoE^−/−^ mice aged 8 weeks and fed an HFD (high-fat diet) for 4 months were used to establish atherosclerosis animal models and had significantly elevated blood lipid indicators (Appendix A) and blood vessel wall thickening (Appendix A), indicating that we have successfully constructed an atherosclerosis model. 

Rap/Fe_3_O_4_@Lipo, Rap/Fe_3_O_4_@VHP-Lipo and normal saline were injected into the tail vein of mice fed an HFD for 2 months. After 24 h and 48 h, under a multimodal small animal imaging system, the fluorescence distribution in mice showed that, at 24 h, the fluorescence of the Rap/Fe_3_O_4_@VHP-Lipo treatment group was enriched in the heart, and, at 48 h, was still enriched in the site of the heart, while the fluorescence of the Rap/Fe_3_O_4_@Lipo treatment mice mainly distributed in the liver and kidney at 24 h and basically was excreted from the body at 48 h (Figure 4a). As we all know, the aorta of the mouse is connected to the heart. In the previously reported articles on targeted nanoparticles targeting to atherosclerotic lesions in vivo, the aorta and heart are often used as target sites [27,34,35]. Our results showed that Rap/Fe_3_O_4_@Lipo and Rap/Fe_3_O_4_@VHP-Lipo had significantly different enrichment sites, suggesting that Rap/Fe_3_O_4_@VHP-Lipo would have a targeting effect to atherosclerotic lesion sites in mice. Afterwards, the mice were euthanized and dissected, and the aortas were immediately measured under a multimodal small animal imaging system. The results showed that the fluorescence intensity values of the aortas in the Rap/Fe_3_O_4_@VHP-Lipo group were significantly stronger, 2.5 times and 3.4 times, than those of the Rap/Fe_3_O_4_@Lipo group and normal-saline-treated group, respectively, indicating that the Rap/Fe_3_O_4_@VHP-Lipo targeted to the site of early atherosclerotic lesions (Figure 4b,c). These also confirmed that our analysis of the previously obtained in vivo fluorescence experimental results (Figure 4a) is correct. 

LC–MS (liquid chromatography–mass spectrometry) is a detection technology that integrates separation and analysis, and it is also a more accurate technical method currently used to measure pharmacokinetic curves because it can detect extremely small amounts of analytes [36]. In order to determine the pharmacokinetic curve of synthesized Rap/Fe_3_O_4_@VHP-Lipo and compare it with free rapamycin and Rap/Fe_3_O_4_@Lipo, LC–MS was used to measure the amount of rapamycin in mice blood at different time intervals after intravenous injecting of the two kinds of liposomes and free rapamycin into ApoE^−/−^ mice, all with a dosage of 2 mg Rap/kg (according to 2 mL liposomes/kg to calculate the injection volume of each mouse). We first explored the chromatographic and mass spectrometric optimization conditions for the detection of Rap. After a 5-min column separation gradient elution, the retention time of Rap in the column was 3.43 min (Appendix A). The mass spectrum showed that the relative molecular mass of the detected Rap was 936.7, and the three characteristic ion series were m/z→607.333, 409.236 and 453.222 (Appendix A); a standard curve was obtained based on these conditions and a linearity equation between rapamycin concentration (X) and peak area (Y) determined by LC–MS: Y = 655.4X − 225.2 (Appendix A). The concentration of Rap in whole blood was measured from 1 h after intravenous injection and continued to 48 h. According to the obtained Rap content, the pharmacokinetic curves of the three drugs in the atherosclerotic mice were found, respectively. The results showed that the blood circulation half-life (T1/2) of Rap in the free Rap group was 8.0 h; in the Rap/Fe_3_O_4_@VHP-Lipo group, it was 13.84 h; in the Rap/Fe_3_O_4_@Lipo group, it was 15 h (Figure 4d). The Tmax value (reaching Cmax) of the Rap/Fe_3_O_4_@Lipo and free Rap group was 1 h, while the Tmax value of the Rap/Fe_3_O_4_@VHP-Lipo group was 3 h. Compared with the free Rap treatment group, the Rap/Fe_3_O_4_@Lipo and Rap/Fe_3_O_4_@VHP-Lipo groups had longer t_1/2_, and the Rap concentrations of blood at 12 h, 24 h and 48 h were significantly higher; the Rap concentrations of the Rap/Fe_3_O_4_@VHP-Lipo groups were significantly higher than those of the Rap/Fe_3_O_4_@Lipo groups throughout the process (Figure 4e). This indicated that the Rap encapsulated in liposomes has a long in vivo circulation time and delayed release of Rap from the targeting group in vivo, especially Rap/Fe_3_O_4_@VHP-Lipo.

### 3.5. MRI Effect of Rap/Fe_3_O_4_@Lipo (No Targeting Peptide) and Rap/Fe_3_O_4_@VHP-Lipo In Vivo and Ex Vivo

After exploring the targeting ability of Rap/Fe_3_O_4_@VHP-Lipo in vivo and in vitro, we followed up by performing MRI function tests. The transverse relaxation rate (r2) is often used to measure the contrast enhancement effect of an MRI contrast agent. Therefore, T2 mapping sequence imaging is used to test liposomes’ MRI effect as this technique should be closer to the true T2 relaxation time. The MRI effect in vitro of liposomes at different concentrations of 0, 10, 20, 30, 40, 60 and 100 μg/mL showed that, as the concentration increased, the T2 relaxation time was gradually decreased, and Rap/Fe_3_O_4_@Lipo has a faster reduction of T2 relaxation time than Rap/Fe_3_O_4_@VHP-Lipo (Figure 5a). Next, by measuring the T2 relaxation time of the three consecutive layers in the middle of the T2 mapping image and taking the reciprocal to generate a linear relationship with concentrations of magnetic liposomes, the slope is r2. We obtained the r2 of Rap/Fe_3_O_4_@Lipo, which was 85.73 M-1s-1, and the r2 of Rap/Fe_3_O_4_@VHP-Lipo was 55.16 M-1s-1 (Figure 5c). These results suggested that, the faster the T2 relaxation time decreased, the greater the r2 value, and both Rap/Fe_3_O_4_@VHP-Lipo and Rap/Fe_3_O_4_@Lipo had good T2 imaging effects. 

In order to detect the targeting effect of Rap/Fe_3_O_4_@VHP-Lipo in early atherosclerotic plaques, the mice aged 8 weeks fed an HFD for 2 months were injected with normal saline, Rap/Fe_3_O_4_@Lipo and Rap/Fe_3_O_4_@VHP-Lipo for 24 h in the tail vein and used for MRI. It was clearly observed that the T2 signal intensity of the carotid artery plaque regions for the Rap/Fe_3_O_4_@VHP-Lipo-treated group mice were negatively enhanced compared to the other two groups (Figure 5b), and the statistical results showed that the Rap/Fe_3_O_4_@VHP-Lipo and Rap/Fe_3_O_4_@Lipo groups, respectively, reduced 2.44 times and 1.95 times in T2 value compared with the normal saline group (Figure 5d). These results confirmed that Rap/Fe_3_O_4_@VHP-Lipo had the shortest T2 relaxation time and the best imaging results for the diagnosis of early atherosclerotic plaques. Therefore, in the next acceptance after completing two months of therapy, we used Rap/Fe_3_O_4_@VHP-Lipo to evaluate the therapeutic effect on atherosclerotic plaques in mice because, the more targeted liposomes aggregate, the more plaques there will be; if the therapy is effective, the plaques will become smaller and the signal will be weaker.

### 3.6. Effect of Rap/Fe_3_O_4_@VHP-Lipo in Therapizing Atherosclerosis

30 mice fed an HFD for 2 months were divided into three groups and therapized with normal saline, Rap/Fe_3_O_4_@Lipo and Rap/Fe_3_O_4_@VHP-Lipo for 2 months. During the 2 months of therapy, the mice continued to be fed an HFD. To determine the therapeutic effect, MRI diagnosis with Rap/Fe_3_O_4_@VHP-Lipo was performed on the mice. The results showed that the carotid arteries of the normal saline and Rap/Fe_3_O_4_@Lipo groups had larger plaque areas and even blocked blood vessels, while the blood vessel wall of the Rap/Fe_3_O_4_@VHP-Lipo therapy group was significantly wider and rounder than the other two groups (Figure 6a). Further, the statistical results showed that the T2 relaxation time in the Rap/Fe_3_O_4_@VHP-Lipo therapy group was 8.9 times and 2.0 times higher than in the normal saline and Rap/Fe_3_O_4_@Lipo therapy groups, respectively (Figure 6b). These indicated that less Rap/Fe_3_O_4_@VHP-Lipo gathered at the plaque in the Rap/Fe_3_O_4_@VHP-Lipo therapy group. A schematic diagram of the animal experimental protocol can be found in Figure 2.

Then, we took blood from the orbit of the mouse to determine the blood lipid levels in the mice. The results showed that, compared with the control group (normal saline), the Rap/Fe_3_O_4_@VHP-Lipo therapy group had a significant reduction in total cholesterol (CHOL) (Figure 6c), triglycerides (TG) (Figure 6d) and low-density lipoprotein cholesterol (LDL-C) levels (Figure 6f); there were no statistical differences in the LDL-C (Figure 6e) and Rap/Fe_3_O_4_@Lipo therapy group; there were significant differences in total CHOL (Figure 6c); there were no statistical differences in the TG (Figure 6d), LDL-C (Figure 6e) and HDL-C levels (Figure 6f). In addition, the mice in all the groups were injected with Rap/Fe_3_O_4_@VHP-Lipo, and, after 24 h, the aortas were dissected and fluorescence images of the isolated aorta results showed that the Rap/Fe_3_O_4_@VHP-Lipo therapy group had the lowest fluorescence intensity compared with the other two groups (Figure 6g). The statistical results showed the fluorescence in the Rap/Fe_3_O_4_@VHP-Lipo therapy group to be 0.57 and 0.68 times lower than that in the control and Rap/Fe_3_O_4_@Lipo groups, respectively (Figure 6h). These results suggested that Rap/Fe_3_O_4_@VHP-Lipo had a very good therapeutic effect and can evaluate the therapeutic effect on atherosclerotic plaques in mice because of its targeting function.

## 4. Discussion

There is much evidence of liposome-linked or encapsulated targeting peptides [37] for the targeting or encapsulation of rapamycin for disease therapy (or that of USPIO), drugs and for early MRI diagnosis and therapy of diseases [38]. Liposomes of fluorescence imaging are also used in the diagnosis and treatment of atherosclerosis [39,40]. However, so far, there are no reports of such comprehensive nanoparticles that we have synthesized by using DSPE-PEG2000-VHP, DSPE-FITC (targeting peptide VHP (or fluorescent reagent) to modify the phospholipid molecules), Rap and encapsulated USPIOs, which target atherosclerotic plaque and bimodal imaging to achieve diagnostic and therapeutic effects for early atherosclerosis. Therefore, Rap/Fe_3_O_4_@VHP-Lipo is a nanoparticle with future clinical applications.

Early diagnosis and treatment of atherosclerosis is an important step in delaying the development of the disease and reducing mortality. Our goal is to synthesize theragnostic nanoparticles for molecular targeted diagnosis and therapy of early atherosclerosis that have the potential for clinical application. Studies have reported that liposomes can be combined with targeting peptides to obtain the ability to target lesions, and non-invasive imaging targeting VCAM-1 can detect the early stages of atherosclerotic inflammation [9]. Studies have reported that VCAM-1 expression increases in the early stages of atherosclerosis [9]. Overexpression of VCAM-1 leads to increased adhesion between monocytes and activated endothelial cells, and monocytes express more inflammatory cytokines to further stimulate endothelial cells to express more VCAM-1 [6]. Therefore, this will create a positive feedback loop that will continuously encourage the development of atherosclerosis. Therefore, application of the VCAM-1 ligand molecule based on this trait in early atherosclerosis is an effective targeting strategy.

Liposome-based multifunctional nanocarriers can coordinate the simultaneous delivery of multiple drugs to achieve collaborative diagnostic and therapeutic integration. There are many studies and reports on the encapsulation of Rap in liposomes. It is true that there are relatively few research reports on the treatment of atherosclerosis. The Rap-loaded immune liposome inhibits endothelial cell proliferation and migration and the expression of inflammatory mediators [41]. This may provide evidence that liposomes loaded with Rap can be used in the therapy of atherosclerosis. According to previous reports, the dosage of 5 to 10 mg (Rap)/kg is relatively large, and a large amount of Rap in the body can cause serious side effects, such as dyslipidemia. Therefore, reducing the Rap dosage to achieve therapeutic effects is a necessary Rap application regimen [42,43]. We injected Rap/Fe_3_O_4_@Lipo and Rap/Fe_3_O_4_@VHP-Lipo into the tail vein of mice to treat atherosclerosis at the amount of 2 mg (Rap)/kg and have shown significant therapeutic effects. Such a dosage is much less than the previously reported ones.

The deficiency of this article is that there is no free Rap therapy group in the therapy experiment to illustrate the therapeutic effect of rapamycin alone. In the pharmacokinetic experiment, we found that the amount of Rap in the blood of the liposome-treated group was much smaller than that of the free Rap group after 1 h, while the dosage of Rap in the liposome synthesis was 1 mg/mL. Our results showed that the encapsulation efficiency of rapamycin was 92.6% (Appendix A). The Rap content in the blood of the liposome-treated group was significantly lower than that in the free-Rap-treated group, so we speculated that the liposomes were rapidly filtered out by the kidney within an hour, which also indicated that the release of Rap from the blood in the liposome-treated groups is relatively low. We noticed that the Rap content in the Rap/Fe_3_O_4_@VHP-Lipo group at 3 h was higher than at 1 h. We speculate that Rap/Fe_3_O_4_@VHP-Lipo progresses from gathering the plaques first and then Rap is released, which means that the most Rap is released from Rap/Fe_3_O_4_@VHP-Lipo within 1 to 3 h in the blood, which implies that Rap/Fe_3_O_4_@VHP-Lipo has targeting properties. We also noticed that the Rap/Fe_3_O_4_@VHP-Lipo group’s blood Rap content in the Rap/Fe_3_O_4_@Lipo group after 6 h was higher than the Rap/Fe_3_O_4_@Lipo group, the free rapamycin group, which further illustrated the prolonged circulation time of Rap/Fe_3_O_4_@VHP-Lipo.

FITC, the fluorescent molecule used in this study, has limitations in in vivo imaging due to its short wavelength, while near-infrared fluorescence (NIRF) is better suited for in vivo fluorescence imaging experiments due to its longer wavelength and higher tissue penetration than FITC. Therefore, we considered using NIRF molecules to modify the nanoparticles for better in vivo fluorescence imaging. We will further investigate the storage conditions of liposomes and improve their stability. Further, in the next study, we will also look more at how the duration of HFD affects atherosclerotic plaque formation. We plan to continue to make our own contributions in the multimodal molecular imaging diagnosis of early atherosclerosis and high-efficiency of nano-therapeutics.

Rap/Fe_3_O_4_@VHP-Lipo for the MRI of plaques at the carotid artery has less respiratory artifacts than the MRI of plaques at the aorta, causing less adverse effects on the judgment of the scan results, and the characteristic imaging of the carotid artery is often used to detect the risk of cardiovascular events [44]. Therefore, we selectively scan the carotid artery when diagnosing atherosclerosis plaques by MRI. Although our targeted imaging agent, Rap/Fe_3_O_4_@VHP-Lipo, had a lower lateral relaxation rate (r2) than Rap/Fe_3_O_4_@Lipo in in vitro imaging experiments (Figure 5a,c), Rap/Fe_3_O_4_@VHP-Lipo showed better imaging and shorter T2 relaxation time in mice imaging tests (Figure 5b,d) indicating that Rap/Fe_3_O_4_@VHP-Lipo aggregated more at the carotid plaque site than Rap/Fe_3_O_4_@Lipo due to the modification with the VHP-targeting peptide, and the aggregation of a large amount of contrast agent resulted in the T2 relaxation time being significantly lower. According to recent reports, the first clinically available 7-T MRI has received a CE Mark, and indocyanine green (ICG) for NIRF imaging has been cleared by the FDA [45]; thus, combining high-resolution MRI with fluorescence imaging may be used in the clinic soon. Therefore, the bimodal imaging combining high-resolution MRI and fluorescence imaging could be used in the clinic soon, which will be of great importance for the early detection of atherosclerotic diseases and the evaluation of treatment effects, and more diseases could benefit in the future.

## 5. Conclusions

We have developed intravenously injectable multifunctional Rap/Fe_3_O_4_@VHP-Lipo magnetic liposomes, which have a good targeting effect on cholesterol plaques for both imaging and therapy in vivo and a prolonged circulation time of the Rap therapeutic agent, which can be used both for the diagnosis as well as early therapy of atherosclerotic lesions.

The targeting effects of targeted liposomes Rap/Fe_3_O_4_@VHP-Lipo on MAECs overexpressing VCAM-1 were detected by flow cytometry and confocal microscopy. Later, MRI-fluorescence bimodal imaging was used to explore the targeted imaging effects of liposomes in ApoE^−/−^ mice. The results of MRI imaging and fluorescence imaging showed that the T2 relaxation time of the Rap/Fe_3_O_4_@VHP-Lipo group was reduced by 2.7 times and 1.5 times, and the fluorescence intensity increased by 3.4 times and 2.5 times, respectively, compared with the normal saline group and the control liposome treatment group. It showed that Rap/Fe_3_O_4_@VHP-Lipo realized the diagnosis of early AS.

After two months of treatment, the mice in different drug treatment groups were injected with the targeted diagnostic agent, Rap/Fe_3_O_4_@VHP-Lipo, and the treatment effect was evaluated by fluorescence imaging and MRI imaging. The results showed that, compared with the normal saline and control liposomes treatment group, the aortic fluorescence intensity of the Rap/Fe_3_O_4_@VHP-Lipo treatment group was significantly weaker, and the T2 relaxation time was prolonged by 8.9 times and 2.0 times, indicating that the targeted diagnostic agent detected the least plaques in the Rap/Fe_3_O_4_@VHP-Lipo treatment group. The blood lipid index measurement showed the same therapeutic effects. Compared with the normal saline group, the TG, CHOL and LDL-C levels were reduced by 18.6 times, 3.13 times and 2.26 times, respectively, in the Rap/Fe_3_O_4_@VHP-Lipo treatment group. Additionally, compared with the control liposome group, the CHOL levels of the Rap/Fe_3_O_4_@VHP-Lipo treatment group were reduced by 1.9 times. It showed that Rap/Fe_3_O_4_@VHP-Lipo can play a significant role in the treatment of early AS.

In the future, we would like to replace FITC-connected phospholipid molecules with phospholipid molecules connected to red fluorescent reagents (such as cy5.5), which is expected to be used in clinical practice. The red fluorescent phospholipid molecules and fluorescent molecules with targeting peptides can be mixed with stem-cell-derived exosomes and used to trace angiogenesis and evaluate the therapeutic effect.

## Data Availability

Not applicable.

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
