# Peer review of "Targeting Peptide, Fluorescent Reagent Modified Magnetic Liposomes Coated with Rapamycin Target Early Atherosclerotic Plaque and Therapy"

_pharmaceutics, 2022, doi:10.3390/pharmaceutics14051083_

Round 1
Reviewer 1 Report
Dear Authors,
Thank you for submitting your work on nanoparticle therapies for treating atherosclerotic plaques. I found your article well written and appreciate the effort to create effective, helpful, instructive figures. Toxicity studies, MR imaging, and FL imaging all contribute to the paper's convincing results. Only,small edits suggested, see below.
A few comments:
-- Authors state that no free rap treatment is used, however figure 4d and 4e contains "Free Rapamycin" curves. This is confusing.
-- A map or diagram detailing what happened to all 30 mice would be helpful. Treatments, post-sacrifice experimental data collected, etc.
Author Response
Response to Reviewer 1 Comments
A few comments:
-- Authors state that no free rap treatment is used, however figure 4d and 4e contains "Free Rapamycin" curves. This is confusing.
Response 1: Thanks for your valuable suggestion. We mentioned the use of free rapamycin in the main text (3.4. Targeting effect and prolonged circulation time of Rap/Fe3O4@VHP-Lipo in vivo). LC-MS simultaneously detects the pharmacokinetic profiles of free rapamycin and two liposomes in atherosclerotic mice. We wanted to show that Rap encapsulated in liposomes has a long in vivo circulation time and delayed release of Rap from the targeting group in vivo, especially Rap/ Fe3O4@VHP-Lipo.
-- A map or diagram detailing what happened to all 30 mice would be helpful. Treatments, post-sacrifice experimental data collected, etc.
Response 2: Thank you very much for your professional advice. We have added the figure in Scheme 2.

Reviewer 2 Report
The manuscript „VHPKQHR-modified Magnetic Liposomes coated with Rapamycin target Early atherosclerotic plaque and Therapy is a very interesting original research study on targeted drug delivery. The manuscript show balanced references and is generally well described from the experimental part. Moreover, the discussion is well built. I recommend publishing after a few minor comments are addressed:
There are multiple typos (eg. Line 70 nanomparticles) in the text which should be fixed
The following sentence in the introduction needs a reference: “The main methods used in the diagnosis of atherosclerosis are computed tomography (CT), positron emission tomography (PET), ultrasound and magnetic resonance imaging (MRI).”
It is not clear why you use different decimals for multile measurements. Especially 3 decimals for the masses from mass spectroscopy are not necessary.
The Conclusion is very short. Here you should summarize your most important findings and also give a brief outlook.
Author Response
Response to Reviewer 2 Comments
I recommend publishing after a few minor comments are addressed:
There are multiple typos (eg. Line 70 nanomparticles) in the text which should be fixed
Response 1: Thanks for your valuable suggestion. We corrected the above mistakes, and checked and corrected some other mistakes carefully in the manuscript.
The following sentence in the introduction needs a reference: “The main methods used in the diagnosis of atherosclerosis are computed tomography (CT), positron emission tomography (PET), ultrasound and magnetic resonance imaging (MRI).”
Response 2: Thanks for your valuable suggestion. We have added the relevant references.
It is not clear why you use different decimals for multile measurements. Especially 3 decimals for the masses from mass spectroscopy are not necessary.
Response 3: Thanks for your valuable suggestion. The article is a description of the principle of mass spectrometry, not the results of multiple measurements. “The mass spectrum showed that the relative molecular mass of the detected Rap was 936.7, and the three characteristic ion series were m/z→607.333, 409.236, and 453.222 (Figure S4b)” This passage explains that m/z→607.333, 409.236 and 453.222 are the three characteristics of Rap with a molecular weight of 936.7. The instrument will calculate the amount of Rap based on this characteristic.
The Conclusion is very short. Here you should summarize your most important findings and also give a brief outlook.
Response 4: Thank you very much for your professional advice. We rewrote our conclusions and added a brief outlook.

Reviewer 3 Report
Proposed paper constitutes a really interesting, valuable and well-prepared work. It is a really good piece of paper showing an interesting research subject and an extensive experimental part. The only suggested minor revisions concern mainly some editorial aspects as well as improvements of some Figures. All suggestions are described in more detail below:
- Title of the paper should be changed to be clearer and more understandable.
- The novelty of the research should be emphasized in Abstract of the paper. Moreover, Abstract should also be supplemented with some quantified data from performed experiments.
- From editorial viewpoint, the reference in brackets should be placed before the dot in the sentence, not after this punctuation mark. Next, the manuscript should be written in the passive voice - i.e. e.g. instead of "we dissolved" it should be "sth was/has been dissolved".
- The explanation of the abbreviation BSA is missing. In general, paper contains numerous abbreviations thus it is suggested to add additional subsection with all abbreviations and their explanations.
- In one sentence Authors use phrase "12 hours", in another one - 12 h. The notation should be unified.
- Subsection 2.2.2., line 138: phrase "the reaction is stirred at room temperature" should be re-written. The reaction mixture is stirred, not a reaction - this mental shortcut should be corrected to be more precise.
- Line 162: the notation of H2O should be supplemented with subscript.
- Section 2.4.: lines 168-170 should be re-written.
- Section 2.5.: the procedure of viability assay should be described in more detail. Was MTT assay used or what type of the assay?
- Figures, Tables and Schemes should be directly in the subsection related to the specific analysis/data etc. and not in a separate subsection.
- Captions of Figures should be shortened. Furthermore:
- Figure 1e) in the description of y axis "Absorbauce" should be replaced by "Absorbance".
- Figure 2a) - there is no unit of the cell viability.
- Figure 3: units should be added in each axis; additionally Figure 3e) should be enlarged - now it is poorly visible.
- Figure 4c): there is no unit of the fluorescence intensity.
- Figure 5d): there is description of x axis.
- Figure 6: units and descriptions should be added in each axis.
- Final Conclusions should be supplemented with some quantified data.
Author Response
Response to Reviewer 3 Comments
Proposed paper constitutes a really interesting, valuable and well-prepared work. It is a really good piece of paper showing an interesting research subject and an extensive experimental part. The only suggested minor revisions concern mainly some editorial aspects as well as improvements of some Figures. All suggestions are described in more detail below:
Title of the paper should be changed to be clearer and more understandable.
Response 1: Thanks for your valuable suggestion. We have revised the title of the article.
The novelty of the research should be emphasized in Abstract of the paper. Moreover, Abstract should also be supplemented with some quantified data from performed experiments.
Response 2: Thanks for your valuable suggestion. We revised the article summary and added some quantitative data
From editorial viewpoint, the reference in brackets should be placed before the dot in the sentence, not after this punctuation mark. Next, the manuscript should be written in the passive voice - i.e. e.g. instead of "we dissolved" it should be "sth was/has been dissolved".
Response 3: Thanks for your valuable suggestion. We corrected the above mistakes, and checked and corrected some other mistakes carefully in the manuscript.
The explanation of the abbreviation BSA is missing. In general, paper contains numerous abbreviations thus it is suggested to add additional subsection with all abbreviations and their explanations.
Response 4: Thanks for your valuable suggestion. We have added all abbreviations and their explanations to the additional subsection.
In one sentence Authors use phrase "12 hours", in another one - 12 h. The notation should be unified.
Response 5: Thanks for your valuable suggestion. We have unified the notation.
Subsection 2.2.2., line 138: phrase "the reaction is stirred at room temperature" should be re-written. The reaction mixture is stirred, not a reaction - this mental shortcut should be corrected to be more precise.
Response 6: Thank you very much for your professional advice. We rewrote the description of this paragraph.
Line 162: the notation of H2O should be supplemented with subscript.
Response 7: Thanks for your valuable suggestion. We corrected the above mistakes.
Section 2.4.: lines 168-170 should be re-written.
Response 8: Thank you very much for your professional advice. We rewrote the description of this paragraph.
Section 2.5.: the procedure of viability assay should be described in more detail. Was MTT assay used or what type of the assay?
Response 9: Thank you very much for your professional advice. We measured cell viability with CCK-8.
Figures, Tables and Schemes should be directly in the subsection related to the specific analysis/data etc. and not in a separate subsection.
Response 10: Thanks for your valuable suggestion. We corrected the above mistakes
Captions of Figures should be shortened. Furthermore:
Response 11: Thanks for your valuable suggestion. We shortened the captions of some figure.
Figure 1e) in the description of y axis "Absorbauce" should be replaced by "Absorbance".
Response 12: Thanks for your valuable suggestion. We corrected the above mistakes
Figure 2a) - there is no unit of the cell viability.
Response 13: Thanks for your valuable suggestion. The cell viability value generally has no units as it is the ratio of two OD values.
Figure 3: units should be added in each axis; additionally Figure 3e) should be enlarged - now it is poorly visible.
Response 14: Thanks for your valuable suggestion. We enlarged Figure 3e to make it easier to see. The abscissa represents mean fluorescence intensity detected by flow cytometry, which is a relative value and has no unit.
Figure 4c): there is no unit of the fluorescence intensity.
Response 15: Thanks for your valuable suggestion. We corrected the above mistakes
Figure 5d): there is description of x axis.
Response 16: Thanks for your valuable suggestion. The x-axis simply represents the different treatment groups. (Figure 5d) This was indicated by a different colored callout.
Figure 6: units and descriptions should be added in each axis.
Response 17: Thanks for your valuable suggestion. The x-axis simply represents the different treatment groups. (Figure 6b, c,d,e,f,h)This was indicated by a different colored callout.
Final Conclusions should be supplemented with some quantified data.
Response 18: Thank you very much for your professional advice. We rewrote our conclusions.
